The use and evaluation of self-regulation techniques can predict health goal attainment in adults: an explorative study

Plaete Jolien 1 jolien.plaete@ugent.be
De Bourdeaudhuij Ilse 1
Verloigne Maite 1
Crombez Geert 2
1 Department of Movement and Sports Sciences, Ghent University , Ghent , Belgium
2 Department of Experimental Clinical and Health Psychology, Ghent University , Ghent , Belgium
Patton Bob
Electronic publication date: 2016 Feb 4
Publication date: 2016
Volume: 4
Electronic Location ID: e1666
Received 2015 Nov 10; Accepted 2016 Jan 17
Copyright: ©2016 Plaete et al.
Copyright year: 2016
Copyright holder: Plaete et al.
License: This is an open access article distributed under the terms of the Creative Commons Attribution License, which permits unrestricted use, distribution, reproduction and adaptation in any medium and for any purpose provided that it is properly attributed. For attribution, the original author(s), title, publication source (PeerJ) and either DOI or URL of the article must be cited.
License URL: https://creativecommons.org/licenses/by/4.0/

Keywords: Self-regulation, Implementation intentions, Plan quality, Health goal attainment, Action planning, Fruit intake, Physical activity, Vegetable intake, eHealth

Funding: Ministry of the Flemish Community, Department of Welfare, Public Health and Family Research Foundation Flanders FWO13/PDO/191 The study was commissioned, financed and steered by the Ministry of the Flemish Community, Department of Welfare, Public Health and Family. Maïté Verloigne is supported by the Research Foundation Flanders (FWO) (postdoctoral research fellowship: FWO13/PDO/191). The funders had no role in study design, data collection and analysis, decision to publish, or preparation of the manuscript.

==============================
Background. Self-regulation tools are not always used optimally, and implementation intention plans often lack quality. Therefore, this study explored participants’ use and evaluation of self-regulation techniques and their impact on goal attainment.

Methods. Data were obtained from 452 adults in a proof of concept (POC) intervention of ‘MyPlan’, an eHealth intervention using self-regulation techniques to promote three healthy behaviours (physical activity (PA), fruit intake, or vegetable intake). Participants applied self-regulation techniques to a self-selected health behaviour, and evaluated the self-regulation techniques. The quality of implementation intentions was rated by the authors as a function of instrumentality (instrumental and non-instrumental) and specificity (non-specific and medium to highly specific). Logistic regression analyses were conducted to predict goal attainment.

Results. Goal attainment was significantly predicted by the motivational value of the personal advice (OR:1.86), by the specificity of the implementation intentions (OR:3.5), by the motivational value of the action plan (OR:1.86), and by making a new action plan at follow-up (OR:4.10). Interaction-effects with behaviour showed that the specificity score of the implementation intention plans (OR:4.59), the motivational value of the personal advice (OR:2.38), selecting hindering factors and solutions(OR:2.00) and making a new action plan at follow-up (OR:7.54) were predictive of goal attainment only for fruit or vegetable intake. Also, when participants in the fruit and vegetable group made more than three plans, they were more likely to attain their goal (OR:1.73), whereas the reverse was the case in the PA group (OR:0.34).

Discussion. The chance that adults reach fruit and vegetable goals can be increased by including motivating personal advice, self-formulated action plans, and instructions/strategies to make specific implementation intentions into eHealth interventions. To increase the chance that adults reach short-term PA goals, it is suggested to keep eHealth PA interventions simple and focus only on developing a few implementation intentions. However, more research is needed to identify behaviour change techniques that can increase health goal attainment at long-term.

Introduction

Physical activity (PA) and a varied diet with fruits and vegetables are associated with decreased risk of cardiovascular diseases and cancer (Lock et al., 2005; WHO, 2009; WHO, 2010). Therefore, adults are recommended to perform at least 30 min of PA at moderate to vigorous intensity on most, preferably all days of the week, and to consume at least 400 g of fruit and vegetable per day (Haskell et al., 2007). However, many adults do not meet these recommendations (WHO, 2003). Despite the efforts to promote these health behaviours in adults, fruit and vegetable intake have been decreasing, and PA levels have remained the same since 2008 in Belgium. A recent meta-analysis focusing on these health behaviours indeed stated that changing unhealthy lifestyle is difficult, and that there is room for improvement (Hallal et al., 2012). In previous computer-tailored interventions grounded in social-cognitive theories (e.g., Theory of Planned Behaviour), tailored feedback was given on motivational determinants such as awareness, knowledge, subjective norm and outcome expectations. Based on the individuals’ scores on scales that measure these determinants, participants were provided with feedback that included a number of tips and suggestions for increasing or maintaining health behaviour (De Vries & Brug, 1999; Kroeze, Werkman & Brug, 2006; Vandelanotte, 2003). For example, participants who had a positive attitude regarding PA, but who were not aware that they were not sufficiently physically active, mainly received information about PA norms and on how to increase PA levels. On the other hand, participants who had negative attitudes got tailored feedback on advantages of PA. However, interventions grounded in social-cognitive theories often only target determinants that are important during the early stages of behaviour change. They are also often more effective in changing intentions than in changing behaviour (Hagger et al., 2012; Sheeran et al., 2005), resulting in a so-called intention-behaviour “gap”. This gap can be targeted by also adopting self-regulation techniques. One useful framework in this context is the Health Action Process Approach model that includes both pre-intentional processes that lead to a behavioural intention and post-intentional processes that lead to the actual health behaviour (Schwarzer, 2008). The model states that individuals first have to become conscious of their own health behaviour and have to be become motivated to change their behaviour, whereafter they have to initiate the new health behaviour to bridge the gap between intentions and behaviour. This can be achieved by defining specific action plans about ‘when’, ‘where’, and ‘how’ to perform the health behaviour, and by stating implementation intentions in which strategies to initiate the action are stated (i.e., “If situation Y is encountered, then I will initiate goal-directed behaviour X”) (Gollwitzer & Sheeran, 2006). People may also make coping plans in which they state how to cope with anticipated barriers and problems that may hinder goal attainment (Bélanger-Gravel, Godin & Amireault, 2013; Schwarzer, 2008; Sniehotta, Scholz & Schwarzer, 2006). Research has shown that interventions that applied self-regulation techniques (i.e., specific goal setting, implementation intentions, providing feedback on performance, prompting review of behaviour goals, social support and self-monitoring) were more effective in changing health behaviour than other interventions that only targeted pre-intentional determinants in tailored feedback (Broekhuizen et al., 2012; Lara et al., 2014; Michie et al., 2009; Morrison et al., 2012).

Therefore, based on previous intervention studies (Spittaels, De Bourdeaudhuij & Vandelanotte, 2007; Springvloet, Lechner & Oenema, 2014; Van Genugten et al., 2010; Vandelanotte, 2003) and the meta-analyses of Michie et al. (2009) and Gollwitzer & Sheeran (2006) we integrated different behaviour change techniques into a novel self-regulation eHealth intervention that targets both pre-intentional and post-intentional processes. Pre-intentional processes were targeted with tailored feedback. Post-intentional processes were addressed with action planning, implementation intentions, problem solving, sharing action plans with friends/family for social support, stimulating self-monitoring and goal evaluation and adjustment.

‘MyPlan’ provided the opportunity to select one out of three health behaviours (fruit, vegetables and PA), provided tailored feedback to prompt intention formation, helped adults to set personal goals, and guided them to plan their behaviour and anticipate barriers and hindering situations during goal pursuit. Other studies, that integrated planning tools, have shown that many tools (e.g., action planning, implementation intentions) are used suboptimally by participants (Springvloet, Lechner & Oenema, 2014; Van Genugten, 2011; Van Osch et al., 2010). For example, Michie, Dormandy & Marteau (2004) found that more than one-third of pregnant women intending to undergo prenatal screening did not formulate implementation intentions for attending or making an appointment despite being prompted to do so (Michie, Dormandy & Marteau, 2004). Furthermore, when self-regulation tools were used, participants did not optimally apply them. Van Osch et al. (2010) reported that plans to promote smoking cessation that are relatively broad and non-specific resulted in less successful behavioural change. Ziegelmann, Lippke & Schwarzer (2006) evaluated completeness of fruit and vegetable plans developed by young, middle-aged and older patients in a rehabilitation clinic. They found that plans that were incomplete (lacking action planning or coping planning) were associated with less PA during rehabilitation at 6 months post-test. This shows that self-regulation techniques are perhaps not always feasible, or are difficult to apply. Therefore, it is important to test whether behaviour change techniques that are included in new interventions are acceptable and feasible for the intended target population and to examine the quality of action plans (Tones & Tilford, 2001). The first aim of this study was to evaluate whether the use of several self-regulation techniques (e.g., selecting hindering factors and solutions, monitor behaviour) and feasibility of the self-regulation techniques (e.g., the difficulty experienced when making an action plan, perceived feasibility of the action plan) could predict goal attainment in the target group (i.e., adults over 18 years). Second, we rated the quality of implementation intention plans and evaluated if the total number of instrumental plans and the specificity score of the implementation intention plans could predict goal attainment. Finally, the moderating effect of the selected behaviour (fruit intake, vegetable intake, PA) on the predictions of use and feasibility rating of the self-regulation techniques on goal attainment was examined, as previous research showed that the effect of behaviour change techniques, may vary for different behaviours (Bélanger-Gravel, Godin & Amireault, 2013).

Methods

Participants and procedure

Data were obtained from participants in a proof of concept (POC) intervention of ‘MyPlan’. ‘MyPlan’ provides personal feedback and helps adults to set and monitor personal and attainable health goals in order to increase either PA level, fruit or vegetable intake. Participants were recruited by distributing flyers to parents of adolescents in secondary schools, by using Facebook and Twitter advertisements and by recruiting university students. Eligible participants were over 18 years, were able to understand Dutch, and had access to Internet. Potential participants were invited to visit the website. A computer log in system was used to allocate adults to the control or intervention condition. The present study only used data from participants in the intervention group who applied at least one of the self-regulation tools of ‘MyPlan’. In the ‘MyPlan’ intervention programme, adults themselves chose a health behaviour that they wanted to change (fruit, vegetables or PA), whereafter they filled in online questions about demographic variables (age, gender, socio economic status) and questions about the selected health behaviour. Next, adults had access to the computer-tailored intervention module (T1). After one week (T2) and one month (T3), adults received an email with an invitation for the follow-up modules. These follow-up models evaluated whether they had reached their health goals and whether they attained the recommended health norms. Figure 1 shows the flow of the participants through the intervention modules as a function of the selected health behaviour. The study was approved by the Ghent University Ethics Committee (approval number of the Ghent University Ethics Committee: 670201319313), and an informed consent statement was obtained from each participant.

Figure 1 Flow chart response rate.

‘MyPlan’ intervention

‘MyPlan’ is informed by self-regulation and Health Action Process Approach theory. After logging in at the website (www.mijnactieplan.be), participants selected a behaviour of interest (fruit intake, vegetable intake, or PA) and completed the first module for that behaviour, which consisted of several components.

Tailored feedback is based upon the answers provided on a questionnaire about the selected behaviour. For PA, the International Physical Activity Questionnaire (IPAQ) was used (Vandelanotte et al., 2005). Tailored feedback consisted of reporting the actual level of PA in different domains (i.e., leisure time PA, active transportation, PA at work, house hold PA), providing feedback about these levels taking into account the health norms, and suggestions to increase PA. For fruit intake, the average portion of fruit per day was calculated using the Flemish ‘Fruit Test’. Participants were asked to indicate how many pieces of each type of fruit that they ate during the previous week. The average portion of fruit per day was calculated. Participants received a report of this average portion and a comparison of this portion with the health norms. For vegetables, the average grams of vegetables was calculated by means of the ‘Vegetable Test’ (Plaete et al., unpublished data). Participants were asked to indicate the amount of portions of each type of vegetable they ate during the previous week. The average number of grams of vegetables per day was reported and compared with health norms.

Action plans were formulated by answering a series of questions. For example, participants were asked what they wanted to do (e.g., being more physically active by walking), when they wanted to do this (e.g., every Monday evening), where they wanted to do this (e.g., local park), how long they wanted to do this (e.g., 60 min) and with whom they wanted to do this (e.g., friends). For PA, adults choose in which domain they wanted to increase their PA level (i.e., leisure time, active life style or both), and defined their goal by selecting activities (e.g., walking, swimming, biking) and by indicating the frequency (days per week) and time (minutes per activity) they wanted to spend on the chosen activity. For fruit and for vegetables, participants indicated the number of days and portions of vegetables they wanted to eat.

Next, implementation intentions were stated. Participants were guided to formulate their action plan into an implementation intention plan format (e.g., If it is Monday evening, then I will go to the aerobic lessons in the local gym).

Problem solving was prompted by indicating hindering factors from a predefined list, or–when not listed-by writing down the hindering factors in an open-ended question format. Participants had to reflect upon solutions to overcome these difficulties. This was also done by providing a predefined list of solutions for each hindering factor that could be selected. When not listed, adults could write down their own solutions in an open-ended question format.

Sharing action plans was made possible by providing the participants the opportunity to send their action plan to family or friends for social support.

Self-monitoring of behaviour was prompted by asking adults to keep a record of their PA levels, fruit or vegetable intake by using one of the listed possibilities (i.e., personal paper agenda, mobile phone, Excel sheet, online agenda). When module 1 was finished, adults were also invited by email to report their behaviour on the website. Periodic email reminders were sent to invite adults to fill out a questionnaire about the target behaviour and their goals on the website.

After one week (T2) and one month (T3), participants had access to follow-up modules which assessed whether participants made progress by comparing PA levels/fruit intake/vegetable intake reported at T2/T3 with PA levels/fruit intake/vegetable intake reported at T1. It was also evaluated whether participants reached their goals. Participants could also adapt or maintain their action plan. Action plans could be adapted by stating new goals (easier goals or more difficult goals) and by selecting new difficult situations, hindering factors and solutions. An overview of the intervention programme is given in Fig. 2.

Measures

Demographics

Participants provided information on age, gender and educational level. Participants with a university or college degree were classified as having a ‘high educational level’ whereas participants with a secondary school degree or lower were classified as having a ‘low educational level’. Age was dummy coded into younger adults (≤40 years) and older adults (>40 years).

Figure 2 An overview of the intervention programme.

Outcome variables

Goal attainment at T2 and at T3 was operationalised in terms of whether participants attained at least their goal set at T1.

Use and feasibility evaluation of behaviour change techniques

Participants indicated whether they used particular techniques (selecting hindering factors and solutions, selecting different domains and activities (for PA only), sharing the action plan, monitor behaviour and making a new action plan at T2) (see Table 1). These variables were dummy coded into used (1) or not used (0) (see Table 1).

To evaluate the feasibility of the self-regulation techniques, additional questions were added at the end of the questionnaire in T1. All variables regarding the feasibility of the self-regulation techniques were dummy coded. Table 1 provides an overview of the predictors about the use and feasibility evaluation of the self-regulation techniques.

Table 1 Health behaviour change techniques.

Behaviour change technique	Predictor	Question	Values (dummy coded)	n (%)	
Tailored feedback (feasibility evaluation)	The motivational value of the personal advice	“I think the personal advice is motivating”	Personal advice perceived as motivating (1) Personal advice not perceived as motivating (0)	141 (63.2) 82 (36.8)	
	The awareness of own behaviour	“Did you expect the result of the personal advice in advance?”	Aware of their behaviour (1) Not aware of their behaviour (0)	129 (57.3) 96 (42.7)	
	The instructive value of the personal advice	“I think the personal advice is instructive”	Personal advice perceived as instructive (1) Personal advice not perceived as instructive (0)	142 (63.7) 81 (36.3)	
Problem solving (use)	Selected barriers and hindering situations	“Select those barriers or hindering situations you want to apply or formulate it yourself”	No barriers or hindering situations (0) Selected/formulated barriers or hindering situations (1)	126 (31.3) 277 (68.7)	
Action planning (feasibility evaluation)	Perceived difficulty of making an action plan	“I think it is difficult to make an action plan”	Perceived making an action plan as difficult (1) Perceived making an action plan not as difficult (0)	82 (37.3) 138 (62.7)	
	The motivational value of the action plan	“The action plan motivates me to pursue my goals”	Action plan perceived as motivating (1) Action plan not perceived as motivating (0)	139(62.9) 82 (37.1)	
	The feasibility of the action plan	“My action plan is feasible”	Action plan perceived as feasible (1) Action plan not perceived as feasible (0)	217 (98.2) 4 (1.8)	
(Use)	Selecting different domains for PA	“How do you want to improve you physical activity level?”	By being more active in my free-time (1) By choosing an active life style (0)	99 (54.1) 84 (45.9)	
	Selecting different activities for PA	“ Do you want to select a second activity for your free time plan?”	Yes, I want to perform a second activity (1) No, I do not want to perform a second activity (0)	84 (54.5) 70 (45.5)	
Stimulating self-monitoring (use)	Monitoring behaviour	“Did you monitor your behaviour the past week?”	Did monitor behaviour (1) Did not monitor behaviour (0)	89 (39.6) 136 (60.4)	
Sharing action plan for social support (use)	Sharing the action plan	“Select to share your action plan with friends and family and fill out their email address”	Sent action plan to family/friends (0) Did not sent action plan to family/friends (1)	57 (25.3) 168 (74.7)	
Goal evaluation and adjustment (use)	Making a new plan at T2	Do you want to make a new plan?	Yes, I want to make a new plan (0) No, I want to keep the same plan (1)	28 (20.4) 109 (79.6)	

Quality of implementation plans

Plan quality of implementation intentions (if-then plans) was evaluated by rating instrumentality and specificity of the plan. We used the rating method of Van Osch et al. (2010), rating plans as (1) instrumental or (0) non-instrumental. Plans were rated as instrumental when they were judged to facilitate the chosen behaviour (fruit intake or vegetable intake, PA) and when they were found to be applicable in the situation that was mentioned. The total number of instrumental plans was used for the analysis by dummy coding it into (0) one or two instrumental plans and (1) more than two instrumental plan. Frequent reasons for scoring a plan as not instrumental were nonsense plans, or plans that did not target the chosen behaviour. Non-instrumental plans were not rated for specificity. Specificity was only scored for plans considered instrumental, and was coded as (0) non-specific, (1) medium specific, and (2) highly specific. Non-specific plans were vague plans, which were often applicable to various behaviours (e.g., “When it is Friday, I am going to sport”). Plans that were described with a certain amount of detail and direction, but that were still general and applicable to several actions and/or lacked one of the following elements (when, how long and where) were rated as ‘medium specific’ (e.g., “When I come home after work, I will go playing basket”). Plans were coded as ‘highly specific’ if a sufficient amount of precision and direction of time (Monday evening 8 am) and place (the local swimming pool) was used and if all elements (when, how long and where) were included (e.g., “When it is Monday evening 8 am, I go swimming for 45 min in the local swimming pool”). Participants had the possibility to make several implementation plans. The mean specificity score of all plans was calculated and used in the analysis by dummy coding it into (0) non-specific plans and (1) medium/highly specific plans. Two researchers independently evaluated all plans on instrumentality and specificity. The interrater reliability was high for instrumentality (Cohen’s k0.89) and substantial for specificity (Cohen’s k = 0.76) (Landis & Koch, 1977).

Statistical analyses

Baseline characteristics of participants were analysed using descriptive statistics. Logistic regression analyses were performed to predict whether participants reached their goal (=goal attainment) at T2 and T3. Various predictors were taking into account. These included several self-ratings of the feasibility of the self-regulation techniques: the awareness of own behaviour, the motivational value of the personal advice, the instructive value of the personal advice, the motivational value of the action plan, the feasibility of the action plan and the difficulty experienced when making an action plan. Also, selecting hindering factors and solutions, selecting different domains and activities for PA, sharing the action plan, monitor behaviour and making a new action plan at T2, were added as predictors to take into account the use of these self-regulation techniques. Furthermore, the coded total number of instrumental plans and the mean specificity score of the implementation intention plans were taking into account. All predictors were dummy coded (see Table 1).

First we evaluated whether the evaluation of the self-regulation techniques, use of particular self-regulation techniques, and plan quality of implementation intentions predicted whether health goals were attained across the three groups. Next, interaction terms (predictor × behaviour) were included to investigate whether the predictors of goal attainment differed as a function of the chosen behaviour (‘PA’ or ‘fruit and vegetables’) of participants. Fruit and vegetables were taken together in one category. In case of a significant interaction effect, the estimated predictive main effect of the predictor only applies to the group that was indicated as the reference category (0). For ease of interpretation, we reported odds ratios and confidence intervals for PA indicated as reference category, and for fruit and vegetables indicated as reference category (see Tables 3 and 4). Statistical significance was set at a level of 0.05, p-values between 0.05 and 0.1 were considered borderline significant.

Table 2 Baseline characteristics for the total sample and the three conditions separately.

	Total intervention group (n = 452)	Intervention physical activity (n = 158)	Intervention fruit intake (n = 166)	Intervention vegetable intake (n = 50)	
Age (years)	30.5 ± 12.5	30.5 ± 12.6	28.1 ± 10.9	33.8 ± 13.4	
Gender (% male)	39.2	44.5	47.8	33.3	
Education level (% high university or college)	72.1	73.6	75.8	66.6	
Instrumentality n (%)					
No instrumental plan (N = 6)	6 (1.7)	3 (1.9)	2 (1.4)	1 (2.3)	
One instrumental plan (N = 159)	159 (45.7)	57 (36.3)	60 (40.5)	42 (97.7)	
Two instrumental plans (N = 102)	102 (29.3)	54 (34.3)	48 (32.4)	0 (0)	
Three instrumental plans (N = 68)	68 (19.5)	30 (19.1)	38 (25.7)	0 (0)	
Four instrumental plans (N = 8)	8 (2.3)	8 (5.1)	0 (0)	0 (0)	
Five instrumental plans (N = 3)	3 (0.9)	3 (1.9)	0 (0)	0 (0)	
Six instrumental plans (N = 2)	2 (0.6)	2 (1.3)	0 (0)	0 (0)	
Specificity n (%)					
Low specificity (N = 28)	28 (8.0)	21 (13.0)	3 (2.0)	4 (9.5)	
Medium specificity (N = 219)	219 (62.2)	87 (53.7)	98 (66.2)	34 (81.0)	
High specificity (N = 105)	105 (29.8)	54 (33.3)	47 (31.8)	4 (9.5)	

Table 3 Predicting goal attainment at T2.

	Goal attainment T2 (n = 274)	
Predictor	Main effecta Predictor OR(95%CI)	Interaction effect predictor × behaviour (p-value)	Main effectb Predictor OR(95%CI)	Main effectc Predictor OR(95%CI)	
The motivational value of the personal advice	1.86(1.06–3.27)*	0.090	2.38(1.15–4.94)**	1.16(0.48–2.78)	
The awareness of own behaviour	1.22(0.64–2.31)	0.077	1.65(0.80–3.40)	0.77(0.33–1.76)	
The instructive value of the personal advice	0.89(0.47–1.70)	0.045	1.20(0.59–2.42)	0.49(0.20–1.19)	
Selecting hindering factors and solutions	1.45(0.80–2.65)	0.019	2.00(1.04–3.85)**	0.89(0.43–1.86)	
The coded total number of instrumental plans	0.89(0.52–1.55)	<0.001	1.73(1.02–2.96)*	0.34(0.17–0.64)**	
The mean specificity score of the implementation intention plans	3.50(0.97–12.57)*	0.016	4.59(1.55–13.63)**	2.20(0.71-6.75)	
The difficulty experienced when making an action plan	1.22(0.63–2.34)	0.058	1.68(0.81–3.49)	0.48(0.15–1.60)	
The motivational value of the action plan	1.86(1.06–3.27)*	0.210	2.25(1.08–4.69)**	1.34(0.57–3.13)	
The feasibility of the action plan	1.30(0.62–2.74)	0.516	1.06(0.40–2.81)	1.63(0.59–4.51)	
Sharing the action plan	1.66(0.98–2.79)	0.111	1.97(1.06–3.65)*	1.20(0.46–3.16)	
Notes.

a No interaction term included for behaviour, with fruit and vegetables as reference category (0).

b With included interaction term (predictorXbehaviour), with fruit and vegetables as reference category (0).

c With included interaction term (predictorXbehaviour), with physical activity as reference category (0).

** p < 0.05: significant predictor.

* p < 0.1: borderline significant predictor.

CI confidence interval

OR odds ratio

Table 4 Predicting goal attainment at T3.

	Goal attainment T3 (n = 137)	
Predictor	Main effecta Predictor OR(95%CI)	Interaction effect predictor × behaviour (p-value)	Main effectb Predictor OR(95%CI)	Main effectc Predictor OR(95%CI)	
The motivational value of the personal advice	1.24(0.55,2.78)	0.230	1.52(0.63,3.68)	1.88(0.67,5.30)	
The awareness of own behaviour	1.09(0.49,2.40)	0.188	1.41(0.57,3.45)	0.70(0.26,1.93)	
The instructive value of the personal advice	0.68(0.29,1.59)	0.101	0.38(0.14,1.05)	0.35(0.12,1.04)	
Selecting hindering factors and solutions	0.97(0.44,2.17)	0.019	1.44(0.60,3.47)	0.486(0.18,1.29)	
The coded total number of instrumental plans	0.99(0.46,2.10)	0.003	1.70(0.70,4.11)	0.40(0.16, 1.031)*	
The mean specificity score of the implementation intention plans	1.91(0.41,8.95)	0.035	2.57(0.53,12,41)	1.10(0.22,5.57)	
The difficulty experienced when making an action plan	0.76(0.34,1.69)	0.327	0.458(0.096,2.179)	0,41(0.09,1.78)	
The motivational value of the action plan	1.05(0.46,2.36)	0.228	1.29(0.53,3.17)	0.70(0.25,1.96)	
The feasibility of the action plan	0.66(0.26,1.62)	0.994	0.65(0.20,2.18)	0.66(0.20,2.18)	
Sharing the action plan	1.73(0.74,4.03)	0.243	0.40(0.09,1.86)	0.94(0.26,3.36)	
Monitoring between T1 and T2	1.18(0.57,2.45)	0.618	0.74(0.23,2.39)	0.96(0.32,2.84)	
Making a new action plan at T2	4.10(1.33,12.64)**	0.022	7.54(1.96,28.99)**	1.35(0.34,5.36)	
Notes.

a No interaction term included for behaviour, with fruit and vegetables as reference category (0).

b With included interaction term (predictorXbehaviour), with fruit and vegetables as reference category (0).

c With included interaction term (predictorXbehaviour), with physical activity as reference category (0).

** p < 0.05, significant predictor.

* p < 0.1, borderline significant predictor.

CI confidence interval

OR odds ratio

Results

Baseline characteristics

In the intervention condition, 225 participants started the intervention module for fruit, 84 for vegetables and 267 for PA. Mean age of participants was 30.5 years (SD: 12.5), 39.2% was male and 72.1% had a high educational level. Table 2 presents the baseline characteristics for the sample that completed the intervention programme at baseline (T1). Descriptive percentages regarding the use and evaluation of the behaviour change methods are given in Table 1.

In total, 59% completed module 2 for fruit, 37% for vegetables and 42% for PA. Module 3 for was completed by 36% for fruit, 12% for vegetables and 17% for PA. Logistic regression analysis revealed that older participants (OR = 4.57; 95% CI [2.35–8.91]; p < 0.001) and participants with low education (OR = 1.72; 95% CI [1.06–2.78]; p = 0.028) had a significant higher probability for drop-out at follow-up (T3).

Goal attainment

For all predictors, odds ratios and confidence intervals of the logistic regression analyses are shown in Tables 3 and 4. In what follows, only significant and borderline significant predictors are reported.

Tailored feedback

The motivational value of the tailored feedback was a significant predictor of health goal attainment at T2. There was also a borderline significant interaction-effect with behaviour (p = 0.090), possibly indicating that this only applied for participants in the fruit or vegetable group. Participants in the fruit or vegetable group who perceived the personal advice about fruit or vegetables as motivating were two times more likely to attain their goal at T2 compared to participants in the fruit or vegetable group who did not perceive the personal advice as motivating (OR = 2.38, 95% CI [1.15–4.94]; p = 0.02).

Action planning

Borderline significance was found for the motivational value of the action plan for health goal attainment at T2. Participants who perceived making their action plan as motivating were more likely to attain their goal at T2 than participants who did not perceive this as motivating (OR = 2.25, 95% CI [1.08–4.69]; p = 0.03).

Problem solving

Selecting hindering factors and solutions was a significant predictor, after including the interaction term with behaviour (p = 0.019). Participants in the fruit or vegetable group who selected hindering factors and solutions, were two times more likely to reach their goal at T2 compared to participants in the fruit or vegetable group who did not select hindering factors and solutions (OR = 2.00, 95% CI [1.15–3.47]; p = 0.04).

Implementation intentions

No significant main effects were found for the coded total number of instrumental plans. However, a significant interaction effect was found with behaviour (p < 0.001). Indicating that participants in the fruit or vegetable group who made more than two instrumental implementation plans for fruit or vegetable intake were three times more likely to attain their goals compared to participants in the fruit and vegetable group that made one or two implementation plans (OR = 1.73, 95% CI [1.02–2.96]; p = 0.09). In contrast, participants in the PA group who made one or two instrumental implementation plans for PA were three times more likely to attain their goals compared to participants that made more than two implementation plans for PA (OR = 0.34, 95% CI [0.17–0.64]; p = 0.006). Furthermore, separate analysis solely in the PA group indicated that stating goals in different PA domains (e.g., free time and active living style) was also a significant predictor for PA goal attainment. Participants who stated goals in different PA domains (i.e., goals for their free time and goals for an active living style (e.g., at work) and goals for active transport) were less likely to attain their PA goals compared to participants that stated goals for only one PA domain (i.e., goals for their leisure time only or for active transport only) (OR = 8.07, 95% CI [2.20–29.55]; p = 0.002). The amount of activities selected for goals in the free time could also predict PA goal attainment. Participants who chose two different physical activities (e.g., walking and swimming) were less likely to attain their PA goals at T2 compares to participants who chose only one activity (OR = 0.21, 95% CI [0.08–0.59]; p = 0.003). At T3, the coded total number of instrumental plans was a significant predictor of health goal attainment, when adjusting for behaviour (p = 0.003). Participants in the PA group who made more than two instrumental plans for PA were less likely to succeed in their health goal than participants in the PA group who made one or two instrumental plans (OR = 0.40, 95% CI [0.16–1.03]; p = 0.05).

Borderline significance was found for the mean specificity score of the implementation intention plans for goal attainment at T2. However, the significant interaction-effect with behaviour (p = 0.016) indicates that the estimated effect only counts for participants in the fruit or vegetable group. Participants in the fruit or vegetable group who made specific plans were five times more likely to attain their health goal at T2 (OR = 4.59, 95% CI [1.55–13.63]; p < 0.021).

Adapting plans

Stating new goals at T2 was found to be a significant predictor of health goal attainment. However, the significant interaction-effect with behaviour indicates that the estimated effect only counts for participants in the fruit or vegetable group. Participants in the fruit and vegetable group who did not state new health goals at T2 were more likely to attain their health goal at T3 than participants who stated new goals at T2 (OR = 7.54, 95% CI [1.96–28.99]; p = 0.003).

Discussion

The results of this study provide further information on how the design, feasibility and applicability of health promotion interventions can be improved to promote optimal behaviour outcomes for different health behaviours. Based on the results, feasible behaviour change techniques can be identified and the content of self-regulation interventions can be improved by further including and optimizing the different behaviour change techniques that can predict goal attainment.

Our study revealed several significant predictors of health goal attainment. After one week, goal attainment was predicted by the motivational value of the personal advice, the motivational value of the action plan, selecting hindering factors and solutions, the specificity score of the implementation intention plans, the coded total number of instrumental plans and selecting different PA activities. After one month, only not stating a new action plan for fruit and vegetables in the follow-up module and making fewer implementation plans for PA could predict health goal attainment. This implicates that perhaps other behaviour change methods or techniques to apply these methods need to be integrated and tested to predict long-term goal attainment.

Our results also showed that the efficacy of particular behaviour change techniques varies as a function of type of health behaviour. Some predictors were only significant for fruit and vegetable intake, and other predictors only for PA. The estimated effect of the specificity score of the implementation intention plans, the motivational value of the personal advice, selecting hindering factors and solutions and making a new action plan after one week to attain the health goal was only applicable for participants who chose for fruit or vegetable intake and not for those who chose PA. In line with our results, the meta-analysis of Bélanger-Gravel, Godin & Amireault (2013) revealed ‘small-to-medium’ effect size of implementation intentions on PA compared to ‘medium-to-large’ effect sizes reported by Gollwitzer & Sheeran (2006) on a variety of health-related behaviours.

Moreover, there was one predictor (i.e., the coded total number of instrumental plans) that was positively related to goal attainment for one behaviour, and inversely related for the other health behaviour. In our study, only a borderline significant effect was found for the coded total number of instrumental plans on fruit and vegetable intake. However, the results for fruit and vegetable intake, are in line with those of Wiedemann, Lippke & Schwarzer (2012), who found that forming a large number of plans may be more effective in changing fruit intake than forming few plans. Our results imply that ‘the more plans, the better’ cannot be generalised for all health behaviours. Our study showed opposite results for PA goals. The coded total number of instrumental plans was the only significant predictor for PA goal attainment after one month. Participants in the PA group who made more than two instrumental plans for PA were less likely to succeed in their health goal. Therefore, to increase the chance that adults reach long-term PA goals, our results suggest that PA interventions should be kept simple and focus only on developing a few implementation intentions. The study of Wiedemann et al. (2011) also showed that better intervention effects were associated with two rather than three PA action plans. Due to our small sample group, we could not investigate the optimal number of plans. Therefore, future research may further focus on the optimal number of plans for different behaviours, especially for PA interventions.

In our study, we also conducted separate analyses for the PA group, because participants who chose the PA module had also the opportunity to make plans for the different domains. However, we found that participants who stated goals in different PA domains (i.e., goals for their free time and goals for active transport) were less likely to attain their PA goals compared to participants that stated goals for only one PA domain (i.e., goals for their free time only or goals for active transport only). The amount of activities selected also negatively predicted PA goal attainment. Participants who chose two different physical activities (e.g., walking and swimming) were less likely to attain their PA goals at T2 compared to participants who chose only one activity. This could perhaps be attributed to the fact that PA is a rather complex behaviour to change (Bélanger-Gravel, Godin & Amireault, 2013; De Vet et al., 2009). This also shows that the feasibility of PA goals and plans may be important. However, the feasibility of the action plan was not a significant predictor of PA goal attainment. It may be that adults have difficulties to formulate feasible PA plans, due to the complexity of incorporating PA goals (Bélanger-Gravel, Godin & Amireault, 2013; De Vet et al., 2009). Therefore, it would be beneficial to incorporate computerized feedback that gives advice on the feasibility by comparing the current health behaviour with goals (especially for PA). For example, adults who never ran before and who state a plan to run every day for one hour, may better receive feedback about the unfeasibility of their plan. It seems that a small group was already aware that their plans were not feasible, as they indicated this in their evaluation. It may make no sense to pursue such goals, and in such situations adults are probably better prompted to adapt their goals. It may be a good idea to implement an evaluation of the feasibility of plans by participants in eHealth interventions. Another idea is to give participants advice to start with only one or two plans in one PA domain, and to make repeated and/or additional plans after the first goal is achieved.

Making implementation intentions of medium to high quality predicted goal attainment. The mean specificity score of the implementation intention plans could only predict goal attainment at short-term (at T2, after one week) in the fruit and vegetable group. In our study, implementation intentions were used to let adults make action plans. Bélanger-Gravel, Godin & Amireault (2013) stated that using implementation intentions for PA only for action planning and not for coping planning (i.e., management of barriers) can decrease the efficacy of implementation intentions. This may explain why the use of implementation intentions could not predict goal achievement for PA goals and after a longer period. We did let participants select difficulties/barriers/hindering factors and solutions (i.e., problem solving) but this was not applied in an implementation intention format (i.e., if-then plans) and could also only predict goal attainment in the fruit and vegetable group at short-term. By using the implementation intention format, critical cues in coping plans are linked to the goal-directed behaviour, which creates a strong and automatic cue-response. Previous studies observed the ‘if-then’ format to yield better behaviour outcomes (Armitage & Arden, 2010; Chapman, Armitage & Norman, 2009). Thus, our results strengthen the suggestion of Bélanger-Gravel, Godin & Amireault (2013) to incorporate coping plans into an implementation intention format.

Only a small group sent their action plan for social support as part of ‘MyPlan’, and this could not predict goal attainment. This result shows that further investigation on how to include social support in eHealth interventions is warranted. Morrison et al. (2012) reported in their review study that social context and support mediates eHealth intervention outcomes. To increase intervention effectiveness, they suggest to provide social support by using automated dialogue, peer-to-peer–mediated communication, or information about other real users. Ziegelmann, Lippke & Schwarzer (2006) reported more complete action plans and a longer duration of physical activities when participants were assisted by an interviewer trained in motivational interviewing. This suggests that additional personal support by health counsellors trained in motivational interviewing could also lead to additional effects of future planning interventions (De Vet et al., 2009; Ziegelmann, Lippke & Schwarzer, 2006).

Making a new action plan in the follow-up module was the only significant predictor for fruit and vegetable attainment after one month. Participants who did not state new health goals at T2 were more likely to attain their health goal at T3. This may indicate that the timing and frequency of follow-up modules might be important to attain health goals at the long-term. Adults in the fruit and vegetable group who adapted their plan after one week had less chance to achieve their goals after one month compared to those who did not adapt their plans yet. In the PA group, adapting plans after one week could not predict goal attainment. This indicates that giving people the possibility to adapt their goals after one week is maybe too early. To our knowledge, no studies have investigated the optimal frequency and timing of follow-up modules in self-regulation interventions. Perhaps instructing participants to use follow-up modules at fixed moments is not effective and in contrast to their preference for more flexibility. Therefore, a suggestion for future researchers is to use follow-up modules that are adjustable to the needs of the individual user. Mobile phone apps, for example, make use of real-time assessment, are constantly accessible and can therefore provide data anywhere and anytime. In this way, tailored feedback and follow-up can be provided at the appropriate time and place, adjusted for individual users (Middelweerd et al., 2014). Using smartphone apps also offers the opportunity for users to monitor their behaviour. Michie et al. (2009) showed that self-monitoring of behaviour was associated with improved effectiveness of eHealth interventions. However, in our study, prompting monitoring of self-behaviour could not predict health goal attainment. In MyPlan, participants were prompted to monitor their behaviour by asking adults to keep a record of their PA levels or fruit and vegetable intake by using a proposed suggestions (e.g., in their personal paper agenda). Furthermore, participants could also monitor/track their behaviour by reporting their behaviour on the website, in the follow-up modules. Perhaps, tools like smart phone apps, in which participants can constantly monitor their behaviour at any place and any time and receive constant feedback on their behaviour change progress, will be perceived as more fun and may be more likely to predict health goal attainment.

After one month, only one significant predictor for fruit and vegetable health goal attainment and one for PA goal attainment could be identified. Our results should be interpreted with caution due to the small sample and high attrition rate, which resulted in a restricted statistical power. This may have influenced the results for the impact on goal attainment at T3. After one month (T3) a notable high attrition rate was observed. Our intervention did contain techniques that have been proposed to enhance sustained use (i.e., goal setting, self-monitoring of behaviour). This is a challenge for many computer-tailored or internet interventions (Schneider et al., 2011), and will need to be addressed in order to use the full potential of eHealth interventions. Perhaps, the time needed (e.g., on average 25 min) to complete the first module was too long, or instructions to revisit the website and ways to get access to the follow-up modules were not clear. We only used one email to invite adults to revisit the website. In the future, we may use emails with updated information or an email and SMS reminder system (Schneider et al., 2011). Our drop-out analyses indicated that older participants and participants with lower educational levels had a significant higher probability for drop-out at follow-up (T3). Previous research indicated that participants who complete health interventions tend to be female, middle-aged and high educated (Brouwer et al., 2010; Liang et al., 1999; Schneider, 2013). This argues for a further evaluation of strategies to prevent drop-out, especially in low educated and older adults. Also important is that our study was conducted in a rather young (mean age 30.5 years) and highly educated population, which may have influenced our results. Therefore, we suggest future research to also try to reach other population groups (e.g., older and low educated adults) when testing eHealth interventions. Due to the low power, we also decided to report borderline significant results. Studies with larger samples are needed to confirm our results. Also, the choice options (e.g., choosing to only form plans for PA in leisure time) have led to some small sample groups, making it not possible to perform moderator analyses. Next, the short study duration must be taken into account when interpreting the results. As only two significant predictors for health goal attainment at 1 month follow-up could be found, it is important to further identify behaviour change techniques that can predict health goal attainment at 1 month but also at long-term (e.g., after 6 months and 1 year). Therefore, a longer trial with a larger and robust sample size is needed. Furthermore, it is also important to note that other factors (e.g., quality of theoretical content, combination of behaviour change techniques, participants characteristics) and the combination of factors might also have important effects on intervention outcome and needs further investigation (Michie et al., 2009; Morrison et al., 2012). Finally, future studies should evaluate whether the behaviour change techniques that were theoretically predicted to affect changes in behaviour/health goal attainment can actually influence intervention effectiveness. Therefore, experimental studies with different intervention conditions which do, and do not include sets of behaviour change techniques are needed (Michie et al., 2009).

Conclusion

To increase the probability that adults attain short-term fruit and vegetable goals, we recommend integrating (a) personal advice and self-formulated action plans that are evaluated as motivating by participants, (b) a problem solving tool in which adults can select hindering factors and solutions, (c) the recommendation of making multiple implementation plans, (d) instructions/strategies to make specific implementation intentions. To increase the chance that adults reach short-term PA goals, our results suggest that PA interventions should be kept simple and focus only on developing a few implementation intentions. Furthermore, further evaluation of behaviour change techniques (e.g., use of health behaviour apps for self-monitoring of behaviour and providing real-time follow-up) that can influence long-term goal attainment is necessary.

Supplemental Information

Supplemental Information 1 Dataset goal attainment

Click here for additional data file.

Additional Information and Declarations

Competing Interests

Author Contributions

Human Ethics

Data Availability

The authors declare there are no competing interests.

Jolien Plaete conceived and designed the experiments, performed the experiments, analyzed the data, contributed reagents/materials/analysis tools, wrote the paper, prepared figures and/or tables.

Ilse De Bourdeaudhuij, Maite Verloigne and Geert Crombez conceived and designed the experiments, contributed reagents/materials/analysis tools, wrote the paper, reviewed drafts of the paper.

The following information was supplied relating to ethical approvals (i.e., approving body and any reference numbers):

The study was approved by the Ghent University Ethics Committee (670201319313), and an informed consent statement was obtained from each participant.

The following information was supplied regarding data availability:

The raw data is supplied in the Supplemental Information dataset file.

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
