# Peer review of "The use and evaluation of self-regulation techniques can predict health goal attainment in adults: an explorative study"

_PeerJ, doi:10.7717/peerj.1666_

## Round 0.1 · original submission · Minor Revisions

Please ensure your resubmission is thoroughly proofread. Your discussion should also address the impact at one month vs. longer follow-up, Both reviewers have made a number of suggestions to enhance the paper and you should address each of these.

Reviewer 1 ·

Basic reporting

On page 4, line 66, the authors may consider describing some of these “other” interventions. The self-regulation techniques mentioned cover a lot of ground and it would help the reader to know what other interventions look like.

The paper could use another look for grammar. There are a few places where the verb tense is confusing (notably in the introduction)

Experimental design

It would be useful to the reader to have more information about the prior research on similar interventions and to point out the novelty of the measures used here (if that is the case). Some meta-analyses are mentioned in the introduction, but it’s never spelled out precisely where this prior work falls short or what remaining issues motivated this study. Is it that no prior work has examined the feasibility of people’s strategies for goal attainment? Or is that that no one has examined the quality of implementation intentions?

Validity of the findings

Please list exact p-values.

In the action planning section, it is stated that the motivational value of the action plan was borderline significant but the p-value is listed as less than 0.05.

By my count there are as many as 15 predictors in the regression model. I wouldn’t put much stock in interpreting non-significant effects (between p>0.05 and p<0.01) with that many possible tests.

Is it the case that if an effect occurred at T2 but was not reported for T3 then that effect was specific only to T2? If that is the case, then it seems that the only predictor of goal attainment at one-month (T3) is in the PA group and shows that making fewer implementations intentions is a better predictor of goal attainment than making more implementations intentions. This would undermine most of the conclusions in the discussion and suggest that an intervention focused only on developing a few implementation intentions is the one that would work best for long term goal attainment in the PA domain but not in the food domain. In fact, it appears nothing predicts goal attainment in the food domain at one-month other than not stating new health goals at T2.

From this perspective, the intervention has only minimal effects on long-term/one-month goal attainment and I recommend the authors rework their discussion to address this.

·

Basic reporting

see general comments

Experimental design

no comments

Validity of the findings

no comments

Additional comments

consider starting new paragraph at line 62

can you mention general characteristics about your population sample in your text and perhaps if you believe this affected your findings in any way

can you speak to the attrition in the study

*tracking has also been shown as an important self-regulation technique; can you speak to how participants were informed or instructed on tracking their health behaviors. and if not, can you explain the exclusion of this technique

*it seems there were 3 points of tailored feedback, but that these did not change in any way after the initial sign up? can you clarify if this is the case. if so, can you expand on how more frequent and/or specific feedback could affect perceived motivation and help with formulation of coping strategies.

*perhaps no one study has evaluated optimal frequency & timing of follow-up modules-however, there are numerous studies with varying follow-up methods existing in the literature. this may be helpful in exploring.

*the last two points are worth considering to prevent underestimation of tracking & tailored feedback (type/form & frequency) on action plan execution, coping strategy formulation, and overall goal attainment.

---

## Round 0.2 · accepted · Accept

Thank you for your resubmitted paper. You have clearly described how you have addressed the concerns raised by the reviewers. I am pleased to accept your paper for publication.